# Sensing Magnetic Field and Intermolecular Interactions in Diamagnetic Solution Using Residual Dipolar Couplings of Zephycandidine

**DOI:** 10.3390/ijms232315118

**Published:** 2022-12-01

**Authors:** Radoslaw M. Kowalczyk, Patrick J. Murphy, Jamie Tibble-Howlings

**Affiliations:** 1Chemical Analysis Facility, School of Chemistry, Food and Pharmacy, University of Reading, Whiteknights Campus, P.O. Box 224, Reading RG6 6AD, UK; 2School of Natural Sciences (Chemistry), Bangor University, Bangor LL57 2UW, UK

**Keywords:** residual dipolar couplings (rdc), Nuclear Magnetic Resonance (NMR), zephycandidine, magnetic anisotropy, magnetic susceptibility, aromatic molecule, cohesive energy density (ced)

## Abstract

An unusual residual dipolar coupling of methylene protons was recorded in NMR spectra because aromatic zephycandidine has preferential orientation at the external magnetic field. The observed splitting contains contribution from the dipole–dipole *D*-coupling and the anisotropic component of *J*-coupling. Absolute values of the anisotropy of magnetic susceptibility |Δ*χ_ax_*| are larger for protic solvents because of the hydrogen-bonding compared to aprotic solvents for which polar and dispersion forces are more important. The energy barrier for the reorientation due to hydrogen-bonding is 1.22 kJ/mol in methanol-*d*_4,_ 0.85 kJ/mol in ethanol-*d*_6_ and 0.87 kJ/mol in acetic acid-*d*_6_. In dimethyl sulfoxide-*d*_6_, 1.08 kJ/mol corresponds to the interaction of solvent lone pair electrons with π-electrons of zephycandidine. This energy barrier decreases for acetone-*d*_6_ which has smaller electric dipole moment. In acetonitrile-*d*_3_, there is no energy barrier which suggests solvent ordering around the solute due to the solvent-solvent interactions. The largest absolute values of the magnetic anisotropy are observed for aromatic benezene-*d*_6_ and tolune-*d*_8_ which have their own preferential orientation and enhance the order in the solution. The magnetic anisotropy of “isolated” zephycandidine, not hindered by intermolecular interaction could be estimated from the correlation between Δ*χ_ax_* and cohesion energy density.

## 1. Introduction

Residual dipolar couplings (rdc) have become an important tool in elucidating structure and conformation of macromolecules [1,2,3,4]. They arise because of a direct interaction between nuclear magnetic moments and are observed in solution-state Nuclear Magnetic Resonance (NMR) spectra when free motion of molecules is restricted [5,6]. That is usually achieved by using a diamagnetic medium (such as liquid crystal or polymer gel) with ability to introduce the preferential orientation of solute molecules [1,2,3,4,7,8,9]. Anisotropic interactions are no longer fully averaged in such solution, and they could be experimentally detected and analyzed using Saupe formalism of an alignment tensor and resulting ordering parameter [10,11,12].

Diamagnetic aromatic molecules possess a unique ability to orient themselves spontaneously in the solution in the presence of the external magnetic field without a need for the orientating medium [13,14,15,16,17]. This effect is dependent on the anisotropy of the molecule magnetic susceptibility and is relatively rarely observed experimentally, as the strength of the dipole–dipole interaction depends on the inverse cube of the distance [18,19,20]. Exceptionally stable conditions for a solution-state NMR experiments are required to observe a small dipolar splitting of resonances in the spectrum which are in the order of 1 Hz at the highest achievable magnetic fields (~23 T) for commercially available spectrometers [16]. 

Those difficulties limit the potential use of the residual dipolar couplings to explore the magnetic properties of diamagnetic aromatic molecules in various diamagnetic solvents. In particular, intermolecular interactions between solvent and solute could provide valuable experimental insight into how the isolated diamagnetic molecule senses its environment, how effectively the diamagnetic solvent shields the solute from the external magnetic field and how it controls the molecular dynamic of the solution [21]. 

In this contribution, the rare residual dipolar couplings between methylene protons of zephycandidine at magnetic fields of 16.445, 11.440, 9.390T are reported. Combination of a large magnetic anisotropy of this diamagnetic aromatic molecule and proximity of interacting magnetic moments results in the large splitting (~0.6 Hz) of the resonance line in NMR spectra. This allows elucidation, with significant precision, the effect of the diamagnetic surrounding of the solvent on the magnetic anisotropy of the solute molecule for several deuterated solvents and at the wide range of temperatures. The experimentally estimated magnetic anisotropy clearly depends on the ability of the solvent to suppress fast reorientation dynamic of the solute. Zephycandidine molecule requires additional energy to change its preferential orientation in solvents with the ability to form strong hydrogen-bonds and such additional energy barrier has been observed and estimated for methanol-*d*_4_, ethanol-*d*_6_ and acetic acid-*d*_4_. Dispersion and polar forces dominate interactions for aprotic solvents used. The energy barrier related to the interaction of solvent lone pair electrons with π-electrons of zephycandidine is mediated by the electrostatic forces in dimethyl sulfoxide-*d*_6_ and it is comparable to the energy barrier estimated for hydrogen-bonding. The importance of the molecular dynamic in the solvent-solute system is also confirmed by data which clearly show substantial enhancement of the magnetic anisotropy of zephycandidine dissolved in the aromatic benzene-*d*_6_ and toluene-*d*_8_ which have ability to self-order at the external magnetic field. 

This contribution demonstrates also a unique experimental concept how the geometry of zephycandidine combined with two independent sources of information about its magnetic anisotropy could be used to distinguish between the direct dipole–dipole *D*-coupling and the anisotropic component of indirect *J*-coupling.

This study, for the first time, correlates the magnetic anisotropy of the solute with the experimentally measured cohesion energy density of the solvent. As a result, the anisotropy of the magnetic susceptibility for the solute molecule which does not experience any hindrance for the reorientation in the solution could be estimated.

## 2. Results and Discussion

The ^1^H NMR spectrum of zephycandidine dissolved in methanol-*d*_4_ recorded at 16.445 T (700 MHz spectrometer) is shown in Figure 1. The assignment of all resonances to the molecular structure agrees fully with previously published data [22,23,24].

The resonance at 6.155 ppm shows a splitting of 0.631 Hz which is not expected in solution unless the symmetrical position of methylene protons in the molecule is distorted [25,26] or the motion of the solute is no longer isotropic [13,14,15,16,17]. Experimental conditions must be excluded as a possible source of this splitting because tetramethylsilane (TMS) forms a symmetric Lorentzian-shape resonance [27] with a half-amplitude width of ca. 0.5 Hz (Figure 1). There is also no evidence of any systematic distortions to other resonances in the spectrum (Figure 1). Deviations from the symmetrical position of methylene protons in aromatic zephycandidine could produce inequivalence in the chemical shift, and hence the splitting but would be observed uniformly at both low and high magnetic fields [25,26]. Similarly, an unlikely contribution from the isotropic *J*-coupling would not change between low and high magnetic fields [28,29]. In contrast, the observed splitting decreases to 0.316 Hz at 11.440 T and to 0.209 Hz at 9.390 T as shown in the inset of Figure 1. Therefore, the primary origin of the observed splitting is the anisotropic direct dipole–dipole interaction reintroduced to the NMR spectrum by a partial orientation of solute molecules in the solution [13,14,15,16,17].

Similar results have been obtained for zephycandidine dissolved in several various solvents. Figure 2 shows in more detail, the magnetic field dependence of the splitting for dimethyl sulfoxide-*d*_6_, acetone-*d*_6_, and chloroform-*d*. The additional point was added to the experimental results because any anisotropic component must be equal zero at zero magnetic field (*B*_0_ = 0 T) and in the absence of the isotropic *J*-coupling, in accordance with the quadratic dependence on the external magnetic field *B*_0_ (see Section 3.1 and [18,19,20] for more information). The consistency of the results between various solvents confirms that the experimental conditions or solution preparation are not responsible for the observed splitting because it is highly unlikely for them to influence each investigated sample in the same manner.

The final piece of evidence that the observed splitting of the methylene resonance is caused by anisotropic dipolar interaction is shown in Figure 3(b–b″). This figure presents the magnetic field dependence of the splitting for the resonances which originate from the protons laying in the plane of the molecule. It is clear, that all values increase on decreasing the *B*_0_, which is consistent with the presence of both anisotropic dipolar *D* (dependent on *B*_0_) and isotropic indirect *J*_iso_ (independent from *B*_0_) components to the splitting ∆*υ* (see Section 3.1 and [18,19,20] for more detailed information). 

The observed splitting of the methylene resonance is among the largest observed experimentally because of the proximity of the interacting protons (ca. 1.81 Å) and uniquely its value could be measured directly in the spectrum (e.g., without a need to analyze the differences in *J*-couplings as show in Figure 3(a–a″) for other protons) [19]. This experimental advantage combined with a well-defined molecular geometry of zephycandidine allow to elucidate the anisotropy of the magnetic susceptibility (Δ*χ_ax_*) using simplified Equation (2) as described in Section 3.1 and Section 3.3 and is presented in Figure 2a. Values of Δ*χ_ax_* obtained for various solvents in which zephycandidine fully dissolves are collected in Table 1. The same table also lists the anisotropy Δ*χ_ax_* and rhombicity Δ*χ_rh_* of the magnetic susceptibility estimated in the more conventional way by analyzing *J*-couplings for three- and four-bonds distant protons in the plane of the molecule. The mathematical details of the utilized Equation (4) are given in Section 3.1 and Section 3.3. Figure 3a–a″ shows in detail the resonances selected, and the representative result is presented in Figure 3c. The detailed structural parameters used to estimate Δ*χ_ax_* and Δ*χ_rh_* in the second method are given in Table 2 in Section 3.3.

There are differences between Δ*χ_ax_* estimated using both methods with the second method consistently underestimating the value of Δ*χ_ax_* by ca. 30% (Table 1). There are a few possible reasons which could explain this difference, which is larger than the sum of the experimental errors (see Section 3.3 for details). Firstly, it is possible that the internuclear vector **r***_AA_* (between the methylene protons) is not parallel to the main component of the magnetic susceptibility tensor **χ**. However, the tilt of **r***_AA_* in reference to *χ_zz_* direction should be in the order of ca. 20 degree to account fully for the observed difference. Such tilt is extremely unlikely for a molecule such as zephycandidine because of the geometry of methylene site, axial symmetry, and aromatic character [19,30]. It would also be expected that any deviation from the zero-degree angle would induce an imbalance in the magnetic shielding of the methylene protons and evidence of this should be visible in the NMR spectra consistently at all magnetic fields [25,26]. Secondly, it is necessary to consider that the **r***_AB_* (between the protons in the plane of the molecule) is not perpendicular to *χ_zz_* direction. Again, only tilt of ca. 45 degree would account for the whole difference, and it is rather unlikely that such large distortion is possible [19,30]. The third possibly assumes the presence of the anisotropy of the indirect *J*-coupling. *J*_aniso_ would follow similar dependence on Δ*χ_ax_* and the external magnetic field *B*_0_ as the direct *D*-coupling via its dependence on ordering parameter but has been omitted in the evaluation of mathematical equations (see Section 3.1) because its value is usually negligible for light-nuclei such as protons [31]. Further work would be required to improve experimental data (specially to increase number of magnetic field points), together with detailed calculation to provide more definitive, quantitative conclusion which would separate both contribution to the experimental ∆*υ*. However, it is interesting to point out that according to theory, *J*_aniso_ at the most favorable conditions may contribute up to 30% to the experimentally estimated value of *D* [20].

It is likely that all three factors play some role in the observed difference. However, it is necessary to assume that the *J*_aniso_ is the most significant and therefore, the experimentally measured splitting ∆*υ* of the methylene resonance should be regarded as “effective” value containing both contributions. 

The estimated values of rhombicities Δ*χ_rh_* estimated from the *J*-coupling splitting for three- and four-bonds distant protons in the plane of the molecule are also listed in Table 1. Unfortunately, significant experimental errors prevent any in-depth analysis beyond the qualitative conclusion that they are about two-to-four times smaller than Δ*χ_ax_* estimated in the same procedure for various solvents. That intuitively agrees with the fact that the zephycandidine in-plane structure deviates from a circular geometry and has a more oval elongated shape (see Figure 1) [32].

The experimentally estimated Δ*χ_ax_* are about double the values reported for benzene [16,20] and vary between the solvents with the largest absolute values calculated for aromatic solvents such as toluene-*d*_8_ and benzene-*d*_6_ and the smallest obtained for an aprotic acetonitrile-*d*_3_ and chloroform-*d* (Table 1). Magnetic anisotropies calculated from ∆*υ* of methylene protons have significantly better precision (smaller experimental errors) and only these values are considered in the discussion below.

To understand the differences in Δ*χ_ax_* between solvents, it is necessary to re-examine their role in establishing conditions for the NMR experiments. Figure 4 shows the magnetic anisotropies Δ*χ_ax_* plotted as a function of the cohesion energy density (ced) normalized to the relative magnetic permeability μ_sol_ of the solvent. More details describing the selection of these experimentally measured parameters from previously published data are given in Section 3.4 and Appendix A.

The choice of ced and permeability is related to two major impacts diamagnetic solvent has on the solute molecule: the control of its molecular dynamic and shielding it from the external magnetic field **B**_0_. The ced (equal to energy per molar volume) describes the amount of energy needed to break solvent into separate non-interacting molecules, and intuitively provides information how difficult is to re-arrange molecules in the solvent [33,34]. Such information is relevant because zephycandidine to orient itself in the magnetic field needs to work against the forces which hold solvent together and these exact forces would also help zephycandidine to stay at the preferential orientation before the thermal energy destroys a fragile momentary equilibrium. Relative permeability is very closely related to the volume magnetic susceptibility, i.e., μ_sol_ = 1 + χ_sol_. Magnetic susceptibility has been proven to have a measurable influence on the spectral parameters such as chemical shift and linewidth, and depends on the sample quality, geometry, environment, and temperature [35,36,37,38]. Diamagnetic solvents have a relative magnetic permeability that is less than or equal to one and therefore, the effective magnetic field *B*_eff_ experienced by the solute is usually smaller than the external magnetic field *B*_0_. 

Excluding aromatic benzene-*d*_6_ and toluene-*d*_8_ which special case will be discus later, it is clear from Figure 4, that the absolute values of the magnetic anisotropy |Δ*χ_ax_*| increase with the increase in the ced/μ_sol_ ratio. This suggests that the main reason for the observed differences is the increase in the average time zephycandidine spends at the preferential orientation because the larger the ced/μ_sol_ ratio the more energy required to re-arrange the momentary equilibrium in the solution. This is consistent with the principle of each NMR experiment which provide a snapshot of a time averaged equilibrium and the observation that the magnetic susceptibility could not be a major factor responsible for observed differences because Δ*χ_ax_* scatter randomly as function of 1/μ_sol_ (see inset in Figure 4). The absence of any clear correlation in the inset of Figure 4, also indirectly proves that the experimental error sufficiently accounts for any differences related to the solutions quality and the finite volume and shape of NMR tube between investigated samples.

Despite a clear correlation between Δ*χ_ax_* and ced/μ_sol_, there are two possible inconsistencies: (i) relatively small ced/μ_sol_ ratio in the case of acetic acid-*d*_6_ compared to other protic solvents and (ii) unexpectedly smaller absolute value of the magnetic anisotropy for aprotic acetonitrile-*d*_3_ compared to other aprotic polar solvents (i.e., dimethyl sulfoxide-*d*_6_ and acetone-*d*_6_) (Figure 4).

The reason for a relatively small ced/μ_sol_ ratio in the case of acetic acid compared to other protic solvents (i.e., methanol and ethanol) could be explained by the underestimated experimental value of ced (427 Jcm^−3^), taken from reference [39]. The difficulties of evaluating reliable values of ced for carboxylic acids (acetic acid and formic acid in particular) from thermodynamic experiments are well documented and are related to the presence of both monomers and dimers in the gas phase [40,41,42]. This problem is illustrated in the Appendix A which compares values of ced estimated for acetic acid in various sources [33,39,43,44,45]. The ced calculated from early calorimetric studies (enthalpy of vaporization, Δ*H*_vap_) ranges from 365 Jcm^−3^ for the evaporation of liquid to the equilibrium mixture of gas at saturation pressure, to 858 Jcm^−3^ for liquid to monomer gas experiment [46]. The molecular dynamic simulations reported in [43], also points towards the larger values 694 and 763 Jcm^−3^ (depending on the simulation procedure) and have a fair agreement with 691 Jcm^−3^ calculated directly from Δ*H*_vap_ and reported more recently in [47]. This value is closer to these for ethanol (675 Jcm^−3^) and methanol (874 Jcm^−3^) [39]. Our data presented in Figure 4, could also provide a crude estimate of the ced from the gradient of the linear correlation between the Δ*χ_ax_* and ced/μ_sol_. The value of 911 Jcm^−3^ estimated for acetic acid is larger than that of methanol.

To understand better the differences in Δ*χ_ax_* between solvents and observed inconsistency for acetonitrile-*d*_3_, it is convenient to consider in more detail molecular level interactions which hold each liquid together. To be able to systematically account for the dispersion and polar forces as well as hydrogen-bonding, the total ced could be divided into three components [33,34,39,48]. This approach has been used successfully to predict efficiency of dissolving chemicals in various solvents by a means of Hansen Solubility Parameters (HSP) [48]. The limitation of this analysis is that individual HSP provide only empirical values because it is impossible to separate and measure directly all these interactions in solution. However, the estimated values of HSP are closely related to the physical properties of the molecule, i.e., dispersion to the refractive index and to the boiling point, polar force to the electric dipole moment, whereas the hydrogen-bonding component could be crudely approximated from spectroscopic data or empirically calculated from a molecular structure or total ced [39,48]. It is necessary to clearly stress that the most significant advantage of this empirical approach is that it provides a uniform tool to compare energies of three major interactions for solvents with distinctive properties, which could not be otherwise measured experimentally in a consistent way. 

The effect of polarity on the values of Δ*χ_ax_* is detailed in Figure 5a. There is a trend (within the experimental error) between Δ*χ_ax_* and ced*^P^*/ced for aprotic solvents except for acetonitrile-*d*_3_. The values of |Δ*χ_ax_*| for dimethyl sulfoxide-*d*_6_ and acetone-*d*_6_ are slightly larger than those for chloroform-*d* or tetrahydrofuran-*d*_8_ and that coincide with the difference in the electric dipole moment normalized to molar volume (*p*V_m_^−1^) (Figure 5a). The magnetic anisotropy of zephycandidine in acetonitrile-*d*_3_ is comparable to that for chloroform-*d* (see also Table 1) despite the *p*V_m_^−1^ ratio being much closer to that of dimethyl sulfoxide-*d*_6_ or acetone-*d*_6_ (Figure 5a and Appendix A).

Similar discrepancy for acetonitrile-*d*_3_ could be also seen in Figure 5b which shows Δ*χ_ax_* plotted versus the ced*^D^*/ced ratios. The |Δ*χ_ax_*| decreases slightly for solvents with larger ced*^D^*/ced ratio (or smaller *n*_D_V_m_^−1^) such as tetrahydrofuran-*d*_8_ and chloroform-*d* compared to that for dimethyl sulfoxide-*d*_6_. Such behavior is to be expected because molecules with a large permanent electric dipole will be less sensitive to the presence of weaker dispersive forces but have much greater potential to induce instantaneous dipoles in their environment [49]. That means, that the molecule with the larger permanent electric dipole (i.e., dimethyl sulfoxide-*d*_6_) would have more significant effect on the zephycandidine, whereas molecules with the larger electric polarizability (and smaller permanent dipoles, i.e., chloroform-*d*) could be more influenced by zephycandidine itself.

The reason for the smaller (in the context described above) value of |Δ*χ_ax_*| for acetonitrile-*d*_3_ compared to other aprotic polar solvents (i.e., dimethyl sulfoxide-*d*_6_ and acetone-*d*_6_) is not fully clear. However, it is possible to speculate that this inconsistency could be related to acetonitrile molecules being weakly ordered in the solution as observed experimentally by X-ray diffraction, IR spectroscopy [50] and in the DFT calculations [51,52,53,54]. This would require any solvent-solvent interactions to dominate and effectively suppress solute-solvent interactions (related to electrostatic forces) and would remove any difficulties for zephycandidine to reorient itself in the solution. This is consistent with a relatively small value of the magnetic anisotropy and the absence of the energy barrier for zephycandidine reorientation in the solution due to the solvent-solute interaction (Table 1, see Section 3.5 and paragraph below). The hydrogen bonding is not expected to play major role for acetonitrile-*d*_3_ solution (Figure 5c and Table 1) and the possible error in sample preparation and experimental conditions must be excluded because identical results were obtained for independently prepared samples measured in separated experiments (see Section 3.2).

This situation would be different for any solvent which directly and strongly interacts with the solute. It is clear from Figure 6a, that the splitting Δ*υ* in dimethyl sulfoxide-*d*_6_ do not follow expected by theory dependence on *B*_0_^2^ when compared to that for chloroform-*d* or toluene-*d*_8_ and could not be explained by the temperature dependence of the magnetic susceptibility [36,37,38]. The observed deviation provides a measure of an additional energy which zephycandidine require to reorientation itself in such solution and could be experimentally estimated from the ln(Δ*υ*) plots shown in Figure 6b (more details are given in Section 3.5 and Appendix A). This additional energy barrier of 1.08 kJ/mol is relatively large, despite dimethyl sulfoxide-*d*_6_ being a proton acceptor which is not expected to form strong hydrogen bonds with zephycandidine (Table 1, Figure 5c). However, its oxygen or sulfur atoms could strongly interact with a π-system of zephycandidine and such anion-π and/or cation-π interactions are meditated by electrostatic forces (Figure 5a,b) [55,56]. It is possible that these anion-π interactions dominate in solution because of the axial symmetry of zephycandidine and the geometry of dimethyl sulfoxide molecule. It could be also speculated that the additional energy barrier should be smaller for molecules with smaller than dimethyl sulfoxide-*d*_6_ electric dipole moment and that is the case for acetone-*d*_6_ with the estimated energy barrier of 0.33 kJ/mol (Table 1). 

For protic solvents such as methanol-*d*_4_ and ethanol-*d*_6_, which are proton donors in hydrogen bonds, the estimated values of ∆*E* are 1.22, and 0.85 kJ/mol, respectively (Table 1, Figure 6). It is also clear from Table 1 (and Figure 4), that the magnetic anisotropy for protic solvents is larger than for aprotic solvents. The same can be seen in Figure 5c which shows Δ*χ_ax_* plotted versus ced*^H^*/ced ratio for both protic and aprotic solvents. The correlation between the magnetic anisotropy and the strength of hydrogen-bonding is not surprising. The molecules of zephycandidine are a part of a dynamic system in which constant motions are affected by the presence of external magnetic field **B**_0_ and govern by intermolecular interactions. It is this network of solvent-solute and solvent-solvent hydrogen-bonding which modulates the ability to quickly reorganize the momentary equilibrium, the fingerprint of which is detected in the NMR spectrum [28,29]. 

It is necessary to points out, that in Figure 5c, the ced*^H^*/ced ratio for acetic acid-*d*_6_ is smaller than that of metnanol-*d*_4_ despite nearly identical Δ*χ_ax_* (Table 1) and acetic acid being a stronger proton donor than methanol. This observation mirrors similar discrepancy observed for the ced determined experimentally from thermodynamic studies (see Figure 4 and discussion above) and suggest that experimental difficulties are related to presence of hydrogen-bonded dimers in the gas phase [40,41,42,46]. 

It is clear from Table 1 and Figure 4 that the magnetic anisotropy of zephycandidine in benzene-*d*_6_ and toluene-*d*_8_ has the largest absolute values, despite the evidence of much weaker interactions compared to that present in methanol-*d*_4_ or dimethyl sulfoxide-*d*_6_. However, that is no surprise considering the aromatic character of these solvents, which molecules have themselves a preferential orientation in the external magnetic field [13,15,16,20]. That is also confirmed by the detection of the quadrupolar splitting of ^13^C satellite resonances in their deuterium spectra (Appendix A). This splitting of ca. 0.574 Hz observed in benzene-*d*_6_ agrees with previously reported data at 23.4 T and follows the expected *B*_0_^2^ dependence [16]. There is no significant energy difference for zephycandidine reorientation in neither benzene-*d*_6_ or toluene-*d*_8_ (Figure 6) which confirms that solvent-solute interactions could not cause the increase in the Δ*χ_ax_* values and the preferential orientation of benzene-*d*_6_ and toluene-*d*_8_ is a main mechanism of the observed enhancement. The extent of this enhancement could be crudely estimated by comparing the experimental value of magnetic anisotropy and the value estimated from the correlation gradient in Figure 4, to be in the order of ca. 15%. 

The observed variation in the magnetic anisotropy Δ*χ_ax_* emerge because of significant differences in the reorientation dynamic of zephycandidine in studied solvents which are caused by solvent-distinctive intermolecular interactions. The ced/μ_sol_ ratio provides an adequate measure for such changes in the molecular dynamics because ced describes how difficult is to re-arrange the molecules and μ_sol_ ≈ 1 for diamagnetic solvents. Therefore, the case when ced/μ_sol_ = 0 describes a solute reorientating itself without restrictions imposed by intermolecular interactions with the solvent at unaltered external magnetic field. Considering that, the observed in Figure 4 correlation between Δ*χ_ax_* and ced/μ_sol_ allows to estimate the anisotropy of magnetic susceptibility for an “isolated” zephycandidine molecule not influenced by the solvent to be approximately 1.41 × 10^−27^ JT^−2^.

## 3. Materials and Methods

### 3.1. Mathematical Description of Magnetic Field Induced Self-Orientation

Experimentally detected splitting ∆*υ* of the resonance line in NMR spectrum is a sum of a spin-spin *J*-coupling and dipole–dipole *D*-coupling and is usually express as: ∆*υ* = *J* + 2*D* [20,28,29]. The indirect *J*-coupling could consist of both isotropic and anisotropic contribution, whereas direct *D*-coupling has only an anisotropic term which depends on the orientation of the vector **r***_AB_* linking interacting magnetic moments in respect to the external magnetic field **B**_0_ and magnetic susceptibility tensor **χ**. Using spherical coordinates *D* could be written as [10,11,12,19]:(1)$$D=-\frac{1}{8\pi^{2}}\frac{\mu_{0}\gamma_{A}\gamma_{B}}{r_{AB}^{3}}\frac{B_{0}^{2}}{15kT}\left[\Delta \chi_{ax}\left(3cos^{2}\alpha -1\right)+\frac{3}{2}\Delta \chi_{rh}\left(sin^{2}\alpha cos2\beta \right)\right]$$
where *α* is an angle between the vector **r***_AB_* linking interacting magnetic moments and the *z* axis of **χ**, whereas *β* identifies position of the **r***_AB_* projection onto the *xy* plane of **χ**. In Equation (1), two independent parameters: anisotropy Δ*χ_ax_* and rhombicity Δ*χ_rh_* of diagonal **χ** are used and other symbols have their usual meaning [28,29]. For methylene protons in axially symmetric zephycandidine molecule, there should not be any isotropic *J* contribution to ∆*υ* and a small anisotropic part is neglected as usual in the published literature [20,31]. Further, for planar aromatic molecule such as zephycandidine the principal axis of susceptibility tensor is expected to be normal to the molecule plane [19] and parallel to the **r***_AB_* which allows to express experimentally observed splitting as:(2)$$\Delta \upsilon =-\frac{1}{4\pi^{2}}\frac{\mu_{0}\gamma^{2}}{r^{3}}\frac{B_{0}^{2}}{15kT}\Delta \chi_{ax}$$

From this equation, the anisotropy Δ*χ_ax_* could be estimated, using the magnetic field or temperature dependence of the splitting ∆*υ*. In a similar way, for interacting magnetic moments laying in the plane of the molecule (e.g., the **r***_AB_* is perpendicular to the principal axis of susceptibility tensor), Equation (1) could be rewritten as:(3)$$\Delta \upsilon =J_{iso}-\frac{1}{4\pi^{2}}\frac{\mu_{0}\gamma_{A}\gamma_{B}}{r_{AB}^{3}}\frac{B_{0}^{2}}{15kT}\left[\frac{3}{2}\Delta \chi_{rh}cos2\beta -\Delta \chi_{ax}\right]$$
allowing for independent evaluation of Δ*χ_ax_* and Δ*χ_rh_* which assumes that there is only isotropic contribution of *J*-coupling to ∆*υ* which does not depend on *B*_0_ [28,29].

### 3.2. Sample Solutions and NMR Instrumentation

Zephycandidine powder was synthesized as described in [23]. All solutions were prepared by dissolving zephycandidine powder in high quality deuterated solvents purchased from Sigma-Aldrich (level of deuteration > 99.0 %). The final concentration was 0.020(3) % *w/w* to ensure high level of dilution (solute molecules were isolated). Solutions (0.5 mL) were transferred to high-precision NMR tubes with 5 mm outer diameter and kept at ambient/room temperature conditions. 

The NMR experiments were carried out at 16.445T, 11.745 and 9.390T which corresponds to 700, 500 and 400 MHz Bruker Avance spectrometer, respectively. 700 MHz spectrometer was equipped with a high-sensitivity TCI cryoprobe, whereas standard BBO probes were used for 500 and 400 MHz instruments. The standard procedure of tuning the probe, locking, and shimming was performed for each sample with a minimum 600 s delay after transfer to the superconducting magnet. Spectra were recorded at constant temperature of 297 K or within a range 273 K–336 K equilibrated to 0.2 K accuracy. The minimum spectral resolutions were 0.007, 0.010 and 0.011 Hz and 4, 8 and 8 transients were acquired (1 s relaxation delay) and added together for each 700, 500 and 400 MHz spectrum, respectively. 

### 3.3. Elucidation of RDC from NMR Spectra and Experimental Errors

The experimental values of the methylene protons splitting ∆*υ* were estimated from NMR spectra by: (i) fitting two Lorentzian line shapes and (ii) simulating with the SPINACH package the methylene resonance [57]. The least square method script written in the Matlab was used in both cases [58]. To minimize shimming errors more importance was placed to correctly map the spectral points which amplitude was above the amplitude at half-width than those whose amplitude was below. The staring fitting parameters were allowed to vary at the fixed range until converged. Spectra were simulated with SPINACH simulation package for Matlab using both formalism of the axial ordering matrix and an isotropic *J*-coupling between protons [57]. Due to the speed of simulation the simplified case of isotropic *J*-couplings was used to elucidate all splitting used in subsequent analyzes.

Two independent samples of zephycandidine in methanol-*d*_4_, acetone-*d*_6_ and chloroform-*d* were prepared and measured on all NMR instruments on the same day. These samples were also re-measured 6 months later to account for any instabilities related to the superconductive magnets of the spectrometer. Three to five spectra were recorded for each solution at 297 K, each following the same protocol, from which ∆*υ* were extracted with the standard deviation of 0.015 Hz for 700 MHz spectrometer and 0.025 Hz for 500 and 400 MHz spectrometers. The standard deviation for the non-methylene ∆*υ* was 0.015 Hz for all instruments. 

The anisotropies of magnetic susceptibility (Δ*χ_ax_*) were calculated from the fit (see Figure 2) of the experimental data (including point at 0 T) using the simplified Equation (2) and structural parameters collected in Table 2. That yielded the standard deviation of 0.03 × 10^−27^ JT^−2^. The final experimental error of 0.06 × 10^−27^ JT^−2^ (two times standard deviation; the confidence level 95.4%) for Δ*χ_ax_* was estimated from the methylene protons ∆*υ*. 

For other protons the experimental Δ*χ_ax_* and Δ*χ_rh_* were estimated by extracting first coupling *D* (*D* < 0 in all cases) from the magnetic field dependence of *J* (see Figure 3) and then from the modified Equation (3): (4)$$\frac{-60\pi^{2}r_{AB}^{3}kT}{\mu_{0}\gamma_{A}\gamma_{B}}D_{AB}=\left[\frac{3}{2}\Delta \chi_{rh}cos2\beta -\Delta \chi_{ax}\right]$$

The parameters *r*_AB_ and θ describing the mutual orientation of pairs of interacting protons in the molecule (see Figure 1) are collected in Table 2. The final experimental errors of 0.14 × 10^−27^ JT^−2^ and 0.12 × 10^−27^ JT^−2^ were estimated for Δ*χ_ax_* and Δ*χ_rh_*, respectively. 

**Table 2 ijms-23-15118-t002:** The structural parameters used to elucidate magnetic anisotropy data.

Protons Interacting	$r_{AB}/Å$	θ/deg
methylene	1.81	-
H_2_-H_1_	2.45	0
H_2_-H_3_	2.51	60
H_2_-H_4_	4.18	90
H_3_-H_4_	2.47	30
H_3_-H_1_	4.17	120
H_12_-H_11_	2.76	144

### 3.4. Magnetic Susceptibilities and Cohesion Energy Densities of the Solvents

The values of magnetic susceptibility (χ_sol_) of deuterated solvents in air at normal pressure were taken from reference [38] with exception of acetic acid-*d*_6_ for which the protonated value from [59,60] was assumed in the absence of the value for its deuterated form and ethanol-*d*_6_ which was taken from [61]. Appendix A lists all these values explicitly as well as molar volumes, boiling points, refractive indexes, dielectric constants, and dipole moments for both deuterated and protonated solvents used [62,63]. The values used to calculate the ratio *p*V_m_^−1^ (electric dipole moment to molar volume) and *n*_D_V_m_^−1^ (refractive index to molar volume) in Figure 5a,b, respectively, are also explicitly highlighted in Appendix A. 

The experimental cohesion energy densities (ced/Jcm^−3^) taken from reference [39] were used in Figure 4, whereas empirical ced and ced polar, dispersion and hydrogen bond components were calculated as cube of the Hansen Solubility Parameters from the Appendix 1 in the reference [48]. The differences between these values (calculated for protonated solvents) were assumed to correctly reflect the values for their deuterated versions in the absence of ced’s data for the later. Appendix A also lists for comparison, other experimental and averaged values of ced reported in [33,39,43,44,45,48]. Appendix A compares values of ced for acetic acid from several sources [39,43,46,47,48].

### 3.5. Details of the Energy Barrier Evaluation

The energy barrier ∆*E* was estimated as a difference between the activation energies calculated from the predicted and experimentally measured temperature variations of the ∆*υ* at 16.445 T (Figure 5b).

The previously evaluated magnetic anisotropy ∆*χ_ax_* of each solvent (Table 1) was used to calculate the predicted values of ∆*υ* (from Equation (2) in Section 3.1) at the same temperatures the NMR spectra were recorded. Then, the Arrhenius plot was created for the natural logarithm of the predicted splitting as a function of the inverse of the temperature (dotted lines in Figure 5b). The predicted activation energy was obtained using linear regression fit in Matlab [51].

The same fitting procedure was applied to the experimental variable temperature ∆*υ* data to obtain experimental activation energy and the energy barrier ∆*E* was estimated as a difference between these two values. The experimental error for Δ*E* was estimated to be 0.28 kJ/mol. 

In the Appendix A, the step-by-step description of the procedure above is also given. 

## 4. Conclusions

The aromatic molecules of zephycandidine sense the external magnetic field and have a preferred orientation with respect to its direction. Therefore, they do not rotate freely in the solution but spend considerably more time at a preferred position. This re-introduces an anisotropic interaction and residual dipolar couplings (rdc) are clearly observed in the NMR spectra. 

The contributions of the dipole–dipole *D*-coupling and the anisotropic *J*-coupling to the splitting ∆*υ* of methylene resonance can be quantitatively distinguished because the molecular geometry of zephycandidine makes possible to observe rdc from both methylene protons and protons laying in the symmetry plane of the molecule.

The large anisotropy of the magnetic susceptibility Δ*χ_ax_* and proximity of interacting magnetic moments are responsible for unusually large value of ∆*υ*, and the influence of the diamagnetic surrounding of the solvent on the solute for several deuterated solvents and at the wide range of temperatures could be studied.

The absolute values of Δ*χ_ax_* are larger for protic solvents due to the presence of hydrogen bonds as compared to aprotic solvents for which polar and dispersion forces play more important role in modulating molecular dynamic. The energy barrier attributed to hydrogen bonding is estimated to be 1.22 kJ/mol for methanol-*d*_4_, 0.85 kJ/mol for ethanol-*d*_6_, and 0.87 kJ/mol for acetic acid-*d*_4_. For aprotic but highly polar dimethyl sulfoxide-*d*_6_, the energy barrier of 1.08 kJ/mol corresponds to the direct interaction of the solvent lone pair electrons with π-electrons of zephycandidine. The energy barrier decreases for acetone-*d*_6_ to 0.33 kJ/mol as its electric dipole moment is smaller than that for dimethyl sulfoxide-*d*_6_ and for tetrahydrofuran-*d*_8_ is below the experimental error of 0.28 kJ/mol. For acetonitrile-*d*_3_ no energy barrier has been detected despite its electric dipole moment being comparable to that of dimethyl sulfoxide-*d*_6_. This is consistent with stronger solvent-solvent compared to solute-solvent interaction and could suggest ordering of the solvent molecules around solute. As a result, the fast reorientation of zephycandidine in acetonitrile-*d*_3_, cause the small absolute value of Δ*χ_ax_*. 

The largest absolute values of Δ*χ_ax_* are observed for zephycandidine dissolved in aromatic benezene-*d*_6_ and toluene-*d*_8_ which have their own preferential orientation at the external magnetic field and may enhance the order in the solution-solute system by as much as 15%. 

The experimental cohesion energy density (ced) could be corelated with the changes in the anisotropy of magnetic susceptibility observed for both protic and aprotic solvents and that allows for the first time to approximate the value of Δ*χ_ax_* for an “isolated” zephycandidine molecule, not influenced by the solvent, to be 1.41 × 10^−27^ JT^−2^.

## Figures and Tables

**Figure 1 ijms-23-15118-f001:**
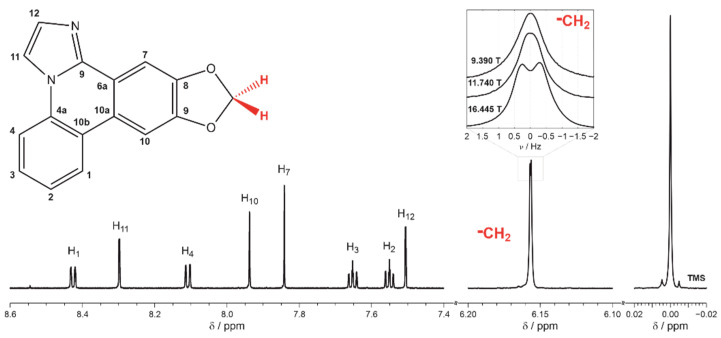
The assignment of ^1^H resonances to the molecular structure of zephycandidine. The methanol-*d*_4_ solution spectrum was recorded at 16.445 T using 700 MHz spectrometer. The inset compares the resonance of methylene protons observed at 16.445, 11.740 and 9.380 T recorded using 700, 500 and 400 MHz NMR instruments, respectively.

**Figure 2 ijms-23-15118-f002:**
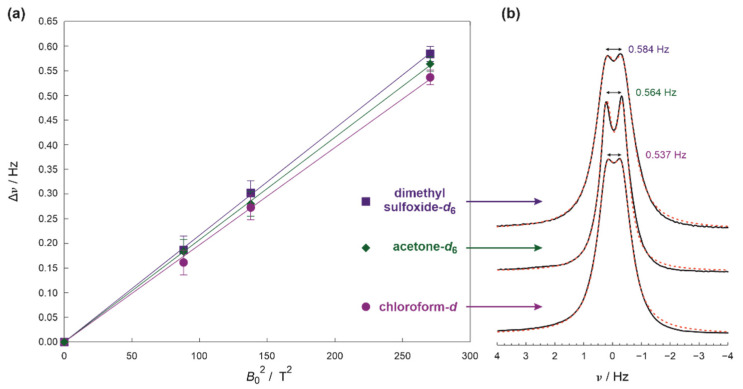
(**a**) The magnetic field dependence of the methylene resonance splitting for zephycandidine dissolved in dimethyl sulfoxide-*d*_6_, acetone-*d*_6_ and chloroform-*d*. (**b**) The methylene resonances (solid black lines) and their simulations (red dotted lines) for these solvents recorded using 700 MHz spectrometer at 16.445 T.

**Figure 3 ijms-23-15118-f003:**
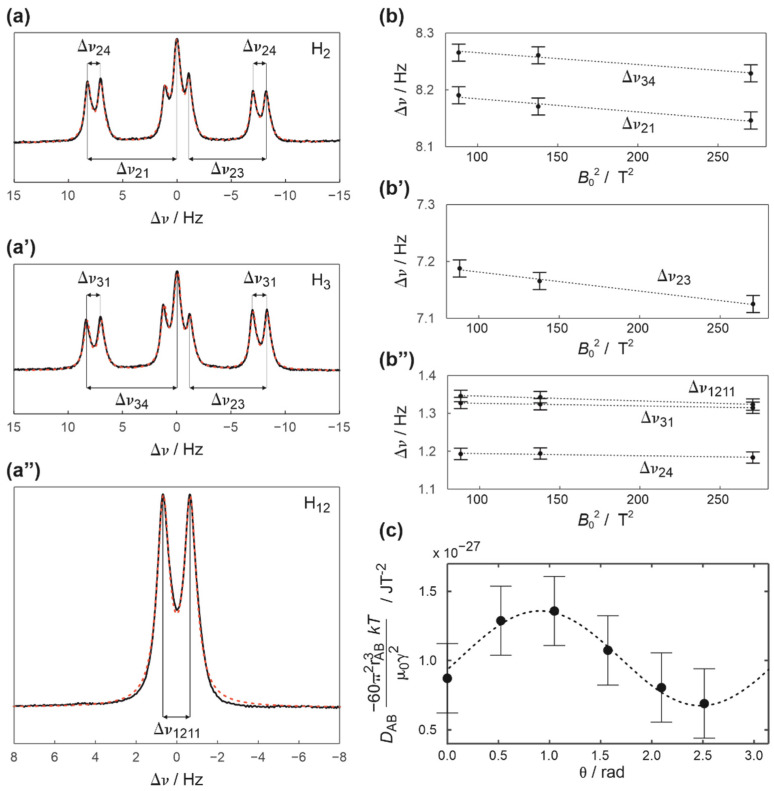
(**a**–**a″**) The resonances of H_2_, H_3_ and H_12_ protons (solid black lines) and their simulations (red dotted lines) obtained for zephycandidine dissolved in tetrahydrofuran-*d*_8_ at 16.445 T using 700 MHz NMR spectrometer. The index to splitting ∆*υ* corresponds to the numbering of interacting protons. (**b**–**b″**) The magnetic field dependence of ∆*υ* used to extract corresponding values of *D*. (**c**) The angular dependence of the normalized values of *D* as detailed in Equation (4) in Section 3.3 used to the estimate magnetic anisotropy parameters Δ*χ_ax_* and Δ*χ_rh_*. The structural details used are listed in Table 2.

**Figure 4 ijms-23-15118-f004:**
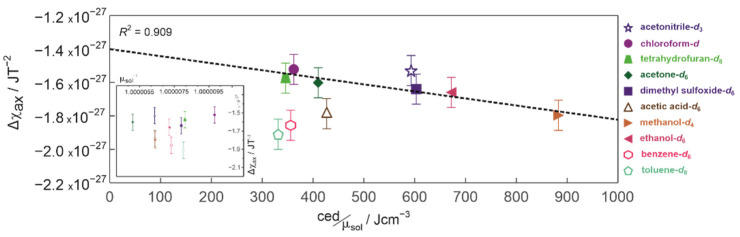
The magnetic anisotropy Δ*χ_ax_* of zephycandidine plotted as a function of the cohesion energy density normalized to the solvent relative permeability. The dotted line represents a linear regression fit to experimental data presented by filled symbols. The inset shows Δ*χ_ax_* as a function of the inverse of magnetic permeability.

**Figure 5 ijms-23-15118-f005:**
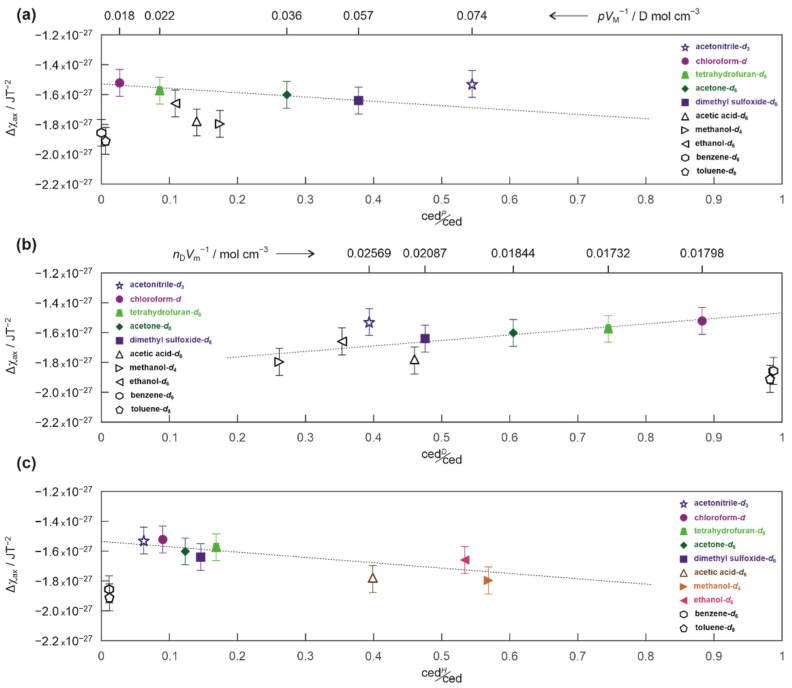
The magnetic anisotropy Δ*χ_ax_* of zephycandidine plotted as a function of (**a**) polar, (**b**) dispersion and (**c**) hydrogen-bonding components of the total cohesion energy density. The dotted lines are for eye guidance only. The experimental data represented by open symbols are not considered for the guideline in particular graph.

**Figure 6 ijms-23-15118-f006:**
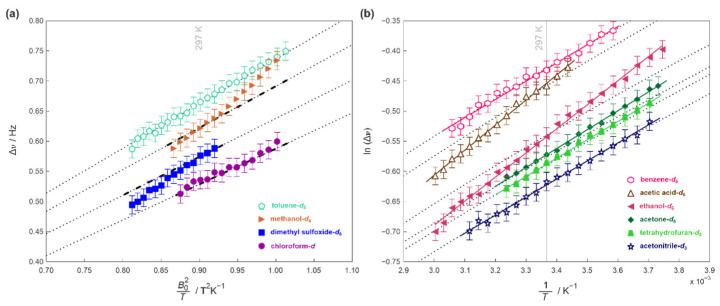
(**a**) The temperature dependence of the methylene rdc splitting recorded at 16.445 T for zephycandidine in selected solvents. The dotted lines show expected theoretical dependence of Δ*υ* on the temperature and the external magnetic field calculated using Equation (2) for given Δ*χ_ax_*. The bold dash lines show the same theoretical dependence but additionally the temperature variation of the magnetic susceptibility of the solvent was included. (**b**) The logarithm of Δ*υ* as a function of the temperature. The solid lines were fitted to the experimental data using the Arrhenius equation. The dotted lines represent expected from Equation (2) theoretical dependence of ln(∆*υ*) on the temperature.

**Table 1 ijms-23-15118-t001:** The anisotropies Δ*χ_ax_* and rhombicities Δ*χ_rh_* of the magnetic susceptibility calculated for zephycandidine dissolved in various solvents and estimated from the residual dipolar couplings of methylene protons (column 3) and in-molecular-plane protons (column 4 and 5). The estimated energy barriers ∆*E* for the reorientation of zephycandidine due to intermolecular interactions are listed in column 6. The experimental errors are given below each column.

	Solvent	$\Delta \chi_{ax}/{\mathrm{JT}}^{-2}$	$\Delta \chi_{ax}/{\mathrm{JT}}^{-2}$	$\Delta \chi_{rh}/{\mathrm{JT}}^{-2}$	ΔE/kJ/mol
aromatic	benzene-*d*_6_	−1.86 × 10^−27^	−1.41 × 10^−27^	−0.35 × 10^−27^	0.00
	toluene-*d*_8_	−1.91 × 10^−27^	−1.51 × 10^−27^	−0.57 × 10^−27^	0.13
protic	acetic acid-*d*_6_	−1.79 × 10^−27^	−1.33 × 10^−27^	−0.41 × 10^−27^	0.87
	methanol-*d*_4_	−1.80 × 10^−27^	−1.32 × 10^−27^	−0.28 × 10^−27^	1.22
	ethanol-*d*_6_	−1.66 × 10^−27^	−1.21 × 10^−27^	−0.29 × 10^−27^	0.85
aprotic	dimethyl sulfoxide-*d*_6_	−1.64 × 10^−27^	−1.21 × 10^−27^	−0.63 × 10^−27^	1.08
	acetone-*d*_6_	−1.60 × 10^−27^	−1.16 × 10^−27^	−0.38 × 10^−27^	0.33
	tetrahydrofuran-*d*_8_	−1.57 × 10^−27^	−1.01 × 10^−27^	−0.21 × 10^−27^	0.13
	acetonitrile-*d*_3_	−1.53 × 10^−27^	−1.01 × 10^−27^	−0.23 × 10^−27^	0.05
	chloroform-*d*	−1.52 × 10^−27^	−1.00 × 10^−27^	−0.27 × 10^−27^	0.02
standard errors (confidence level 95.4%)		0.06 × 10^−27^	0.14 × 10^−27^	0.12 × 10^−27^	0.28

## Data Availability

The data that support the findings of this study are available from the corresponding author upon reasonable request.

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
