# Peer review of "Sensing Magnetic Field and Intermolecular Interactions in Diamagnetic Solution Using Residual Dipolar Couplings of Zephycandidine"

_ijms, 2022, doi:10.3390/ijms232315118_

Round 1
Reviewer 1 Report (Previous Reviewer 3)
The authors improved the previous version of the manuscript and took into account my comments. The paper is worth to be published in IJMS.
Author Response
Please see the attachment.

Reviewer 2 Report (Previous Reviewer 2)
The manuscript is very well written and could be published in its present form with minor corrections as stated below:
- r. 64 - ethanol-d6 instead of entanol-d6
- Conclusion could be shortened, it is too much descriptive.
- The statement is better “Therefore, zephycandidine does not rotate freely in the solution but spends considerably more time at a preferred position” then “Therefore zephycandidine does not rotate freely in the solution but spends considerably more time at this preferred position”.
- r. 593 and 605 have a sign“[“ to be removed.
Author Response
Please see the attachment.

Reviewer 3 Report (Previous Reviewer 1)
The authors have satisfactorily responded to my concerns and made the necessary changes to the manuscript.
Author Response
Please see the attachment.

This manuscript is a resubmission of an earlier submission. The following is a list of the peer review reports and author responses from that submission.
Round 1
Reviewer 1 Report
The manuscript of Kowalczyk et al. reports an experimental study of magnetic field alignment of zephycandidine in organic solvents. The degree of alignment was estimated from the residual dipolar couplings. I have no critical comments on the technical part of the study, the measurement accuracy, and the obtained numerical characteristics of the magnetic susceptibility tensor. It is noteworthy that these characteristics differ for different solvents.
The authors attempted to interpret these changes in terms of the interaction between solvent and solute molecules. The conclusions made by the authors cannot be considered scientifically substantiated. They do not give any justification for why the interaction of a strong proton donor (acetic acid) turned out to be weaker than that of a weak one (methanol). The assumption about the formation of zephycandidine/acetonitrile clusters is based only on the violation of the solvent polarity dependence assumed by the authors. The methodology for calculating the energy barriers for zephycandidine reorientation, which are given in the abstract and conclusion, is not discussed in the article.
Reviewer 2 Report
Very detailed experiments are performed in order to confirm that the observed methylene splitting is the studied compound is caused by anisotropic dipolar and eventually scalar interactions. Many thoughts and calculation are presented to explain the observed facts. Some of them are disputable that deserves the publication in its present form.
Some small editing errors have to be done.
Reviewer 3 Report
The paper “Sensing magnetic field in diamagnetic solution using residual dipolar couplings of zephycandidine” by Radoslaw Kowalczyk et al. concerns very interesting aspect of the restricted motion in high-resolution NMR at various high magnetic fields in different solvents to get information about the magnetic parameters, structure of the complexes, their anisotropy and geometry. The work is well-written and can be accepted for publication. I have only some comments and remarks.
Comments and remarks.
1. Would it be possible to define not only the value but also the sign of the dipolar coupling? What should be done for that?
2. Would it be possible to extract (estimate) the correlation time(s) from the experiments done? What should be done for that? Which type of correlation times used in NMR can adequately describe
3. Would it be possible to measure the influence of the paramagnetic probes (producing large internal magnetic field) by using the proposed in the title technique for sensing huge magnetic fields?
4. Is there any other molecules (systems) except zephycandidine with the same (similar) NMR properties? Or, re-phrased, how to find a series of substances which can be used as magnetic sensors?
5. Some number of typos present in the manuscript which I think can be easily found and corrected.
Round 2
Reviewer 1 Report
Unfortunately, the authors' explanations could not eliminate my doubts. Their conclusions are based solely on correlations with various components of the cohesive energy density, the numerical values of which, by the way, are not given in the article. At the same time, they admit that "it is particularly difficult to evaluate reliable values of ced for carboxylic acids (acetic acid and formic acid in particular) because of experimental complications in evaluating precisely the cedTOTAL from thermodynamic studies (e.g. monomers and dimers heat of vaporization), together with difficulties in estimating of the cedHBOND which accounts for the specificity of hydrogen-bonding". I also don't understand what they mean when they say: “The energy barrier was estimated as a difference between the activation energy calculated for the theoretical temperature dependence of the splitting ΔυT (using equation (2) and experimental value of the magnetic anisotropy) and the directly measured temperature dependence of Δυ at 16.445 T.”
Round 3
Reviewer 1 Report
My doubts are as follows. There is no general method that can be used to measure the energy of specific interactions in a more or less consistent and accurate way for different solvents. In this publication, the authors use the empirical method of Hansen’s solubility parameters without commenting on the limitations of this simplified and purely predictive approach. They then interpret the resulting estimates as "experimental" and draw highly controversial conclusions from them. What is the error in determining the energy of hydrogen bonds in methanol and acetic acid? Is it really only 0.1 kJ/mol, as can be concluded from the article?
Since the use of a rough approximation for a detailed analysis of intermolecular interactions cannot be justified, I consider it necessary to remove all numerical values obtained in the framework of the approach of Hansen’s solubility parameters from the abstract and the summary. These estimates may be given in the text of the article, but it is necessary to caution readers about the estimated nature of these values.
The energy barrier estimate procedure should be explained in the text:
“Step 1. From the magnetic field dependence (equation (2) in the manuscript) at constant room temperature (i.e., 297K) we know the value of the magnetic anisotropy ∆χax for each solvent (listed in Table 1 of the manuscript). Therefore, at a constant magnetic field of 16.445 T (700 MHz NMR spectrometer) we can calculate the “theoretical” splitting ∆υT for other temperatures at that magnetic field from the equation (2).
Step 2. We can fit the natural logarithm of “theoretical” splitting ln(∆υT) as function of the inverse of temperature (T-1) using the Arrhenius equation (red lines in Figure 5(b)). It interesting to point out that the value of this energy is 2.5 kJ/mol for all studied solvents as it is determined by the interaction with the magnetic field.
Step 3. We can fit the natural logarithm of the experimentally measured (at 700 MHz) splitting ln(∆υ) as function of the inverse of temperature (T-1) using the Arrhenius equation. This energy varies between solvents from 2.5 kJ/mol to 3.7 kJ/mol depending on the solvents.
Step 4. The difference in the energies above we called the “energy barrier”. “
